# Tri-Doping of Sol–Gel Synthesized Garnet-Type Oxide Solid-State Electrolyte

**DOI:** 10.3390/mi12020134

**Published:** 2021-01-27

**Authors:** Minji Kim, Gwanhyeon Kim, Heechul Lee

**Affiliations:** Department of Advanced Materials Engineering, Korea Polytechnic University, Gyeonggi 15073, Korea; krmi32@naver.com (M.K.); rhksgus9473@kpu.ac.kr (G.K.)

**Keywords:** solid state electrolyte, all-solid-state battery, lithium-ion battery, garnet oxide, doping, co-doping, tri-doping, sol–gel

## Abstract

The rapidly growing Li-ion battery market has generated considerable demand for Li-ion batteries with improved performance and stability. All-solid-state Li-ion batteries offer promising safety and manufacturing enhancements. Herein, we examine the effect of substitutional doping at three cation sites in garnet-type Li_7_La_3_Zr_2_O_12_ (LLZO) oxide ceramics produced by a sol–gel synthesis technique with the aim of enhancing the properties of solid-state electrolytes for use in all-solid-state Li-ion batteries. Building on the results of mono-doping experiments with different doping elements and sites—Al, Ga, and Ge at the Li^+^ site; Rb at the La^3+^ site; and Ta and Nb at the Zr^4+^ site—we designed co-doped (Ga, Al, or Rb with Nb) and tri-doped (Ga or Al with Rb and Nb) samples by compositional optimization, and achieved a LLZO ceramic with a pure cubic phase, almost no secondary phase, uniform grain structure, and excellent Li-ion conductivity. The findings extend the current literature on the doping of LLZO ceramics and highlight the potential of the sol–gel method for the production of solid-state electrolytes.

## 1. Introduction

The rollout of electric vehicles and increasing interest in energy storage systems has generated rapid growth in the Li-ion battery market. Consequently, there is considerable demand for Li-ion batteries with improved performance and stability [1,2,3]. Conventional Li-ion batteries with liquid electrolytes pose safety risks owing to leakage and volume expansion problems, which can lead to ignition and explosions [4,5,6]. On the contrary, all-solid-state batteries use solid-state electrolytes that eliminate such risks and improve the safety of Li-ion batteries [7,8]. In addition, various battery designs for all-solid-state batteries can be achieved through structural changes, thus enabling improved energy output because they do not require separators [9].

Studies on solid-state electrolytes have considered various approaches and numerous electrolyte materials [10,11,12], including many inorganic materials such as NASICON [13], thio-LISICON [14], perovskites [15], sulfides [16], and garnet oxides [17]. In particular, Li*_x_*La_3_M_2_O_12_ (M = Ta, Nb, Zr) garnet-type oxides have exhibited excellent thermal and chemical stability with a wide electrochemical window and a Li-ion conductivity of ~10^−4^ S/cm [18,19]. Garnet-type Li_7_La_3_Zr_2_O_12_ (LLZO) exists in tetragonal and cubic crystal phases, with the energy barrier against Li^+^ ion migration much lower in the cubic phase [20,21]. Li-ion conductivity can be improved by doping of the Li^+^, La^3+^, and Zr^4+^ cation sites in the LLZO lattice with various dopants to generate vacancies and lattice distortion. This enhances the Li^+^ ion diffusion pathways, and thus increases the ionic conductivity [22,23]. Elements including Al [24], Fe [25], Ta [26], Nb [27], and W [28] have been studied as dopants for LLZO. Most studies employ one or two doping elements that substitute at the Li^+^ and Zr^4+^ sites. However, the effects of doping at the La^3+^ site are still not well understood [29,30,31], and few studies have investigated substitutional doping of all three cation sites [32]. In addition, a conventional solid-state reaction method is typically used [33,34], which presents several limitations including Li loss and secondary phase formation due to the high temperature and prolonged sintering process [35,36].

In this study, we aim to address the limitations of the existing solid-state reaction method by using the sol–gel technique to achieve doping. In addition, we examine the effect of substitutional doping at all three cation sites. Sol–gel based syntheses enable the preparation of uniform nanosized powders that can be used to fabricate solid-state electrolytes with a short sintering time and low temperature [37,38]. We investigated the changes in physical properties according to the dopant content for mono-doping of the Li^+^ site (using Al, Ga, and Ge), La^3+^ site (using Rb), and Zr^4+^ site (using Ta and Nb) based on an optimized LLZO sol–gel synthesis process. Subsequently, considering the results of the mono-doping experiments, we conducted the compositional design of co- and tri-doped LLZO and systematically investigated the properties of doped LLZO solid-state electrolyte ceramics according to the composition. It was found that the properties of the LLZO solid-state electrolyte ceramics were improved by doping with Ga, Rb and Nb at three ion sites. 

## 2. Materials and Methods

### 2.1. LLZO Solid-State Electrolyte Synthesis Using Sol–Gel Technique

The sol–gel process was performed according to a previously published method [39,40]. Prior to this study, optimization of the sol–gel process was conducted to identify suitable processing variables, including the excess Li content (0–40 wt%), calcination temperature (600–900 °C), sintering temperature (800–1100 °C), sintering time (2, 4, or 8 h), and Li source (lithium nitrate hydrate (LiNO_3_∙*x*H_2_O), lithium nitrate (LiNO_3_), or lithium carbonate (Li_2_CO_3_)). LLZO fabrication without secondary phase formation was achieved by the addition of a solid reaction step in which ball milling was performed with the addition of 12 wt% LiNO_3_ (the selected Li source) after calcination.

Lithium nitrate (LiNO_3_, Alfa Aesar), lanthanum nitrate hydrate [La(NO_3_)_3_∙*x*H_2_O, Sigma Aldrich], zirconium(IV) propoxide [Zr(OC_3_H_7_)_4_, 70 wt% in 1-propanol, Sigma Aldrich], tantalum(V) ethoxide (Ta(OC_2_H_5_)_5_, Sigma Aldrich, Darmstadt, Germany), niobium(V) ethoxide (Nb(OCH_2_CH_3_)_5_), Sigma Aldrich, Darmstadt, Germany), aluminum oxide (Al_2_O_3_, 20 wt% in isopropanol, Sigma Aldrich, Darmstadt, Germany), gallium oxide (Ga_2_O_3_, Nilaco Corporation, Tokyo, Japan), germanium dioxide (GeO_2_, Samchun Chemicals, Seoul, Korea), and rubidium carbonate (Rb_2_CO_3_, Sigma Aldrich, Darmstadt, Germany) were used as raw materials for synthesis of the doped LLZO. For effective dissolution, 1-propanol (CH_3_CH_2_CH_2_OH, Sigma Aldrich, Darmstadt, Germany) and 2-methoxyethanol (CH_3_OCH_2_CH_2_OH, Sigma Aldrich, Darmstadt, Germany) were used as solvents. Gelation was achieved by using acetic acid solution (C_2_H_4_O_3_, Sigma Aldrich, Darmstadt, Germany) as a chelating agent.

Li_7.44_La_3_Zr_2_O_12_ was used as the basic composition, to which dopants were introduced at each ion site. The process sequence is shown in Figure 1. The Li and La sources were dissolved in 1-propanol; and the Zr source and Zr^4+^ site dopant (denoted as M, where M = Ta or Nb) were dissolved in 2-methoxyethanol with acetic acid solution. The sol was synthesized by mixing the two solutions at 200 rpm for 45 min. Li^+^ and La^3+^ site dopants (denoted as A and B, respectively, where A = Al, Ga, or Ge and B = Rb) were added to the sol after mixing. The sol was aged for 15 h then gelated by drying at 165 °C for 3 h. The gel was then placed in an alumina crucible for pre-calcination at 450 °C for 4 h. The powder obtained after the pre-calcination process was ground and pressed (8.3 MPa, 60 s) into pellets, followed by calcination in a magnesia crucible at 950 °C for 4 h. After calcination, 12 wt% of the Li source (LiNO_3_) was added, and the powders were ground together by wet ball-milling at 240 rpm for 24 h using zirconia balls and isopropyl alcohol. Drying (100 °C, 3 h) and sieving were performed to obtain nanosized LLZO powder. The powder was pressed at 8.3 MPa for 1 min to form pellets, followed by cold isostatic pressing at 300 MPa for 2 min. The synthesized LLZO solid-state electrolyte ceramic pellets were covered with powder of the same material to prevent Li loss during the heat treatment. A final sinter was performed at 1100 °C for 4 h in a magnesia crucible, followed by air-cooling in the furnace.

### 2.2. Characterization of Solid-State Electrolyte Materials

The fabricated solid-state electrolyte pellets were assessed using high-resolution X-ray diffraction (XRD, SmartLab, Rigaku, Tokyo, Japan) in the 2*θ* range of 10° to 60° to confirm the crystal phase. Field-emission scanning electron microscopy (FE-SEM, Nova NanoSEM 450, ThermoFisher, Waltham, MA, USA) was used to examine the microstructure of the fracture surface of the ceramics. The average grain size was calculated using linear intercept analysis, which was performed on the SEM photographs where ten arbitrary straight lines were drawn. The number of grains was determined for each line, and the grain size was obtained by dividing the line length by the number of grains intersected by the line. The uniformity of grain size was calculated using Equation (1).
*Uniformity*(%) = (1 − *Dev*/*Avg*) × 100,(1)
where *Avg* is the average grain size and *Dev* is the standard deviation of calculated grain sizes. The change in diameter during the sintering process was measured to determine the shrinkage of the LLZO ceramic, and the relative density was calculated using Archimedes’ method.

The ionic conductivity of the solid-state electrolyte was measured using electrochemical impedance spectroscopy (EIS, VersaSTAT 3, Princeton Applied Research, Princeton, NJ, USA). The activation energy was calculated by measuring the ionic conductivity in the temperature range from 20 °C to 80 °C using EIS with a custom heating system. After polishing both sides of the LLZO ceramic pellet to 2000 grit, an ion-blocking capacitor was formed by sputter deposition of 100 nm-thick Au thin-film electrodes on both sides. EIS measurements were conducted at an amplitude of 10 mV in the frequency range of 0.1 Hz to 1 MHz. The impedance was analyzed using the ZView program based on the Nyquist plot obtained through the measurement. The solid-state electrolyte standard equivalent circuit presented by the Telecommunication Technology Association was used. The equivalent circuit, known as Randles circuit is composed of *R*_S_ (the bulk resistance of the solid-state electrolyte), *R_p_* (the interfacial resistance due to charge accumulation), and *CPE* (constant phase element). The ionic conductivity was calculated by substituting the impedance value into Equation (2).
*σ* = (1/*R*_S_)(*t*/*A*),(2)
where *σ* is the ionic conductivity and *A* and *t* are the surface area and thickness of the solid-state electrolyte ceramic pellet, respectively. The activation energy (*E*_a_) was calculated using Equation (3).
*σT* = *B* exp(−*E*_a_/*kT*),(3)
where *B* is a constant, *k* is Boltzmann’s constant, and *T* is the absolute temperature. *E*_a_ was calculated from the slope of the ln(*σT*) graph for 1/*T*, obtained by measuring the ionic conductivity at various temperatures.

## 3. Results and Discussion

### 3.1. Mono-Doped LLZO Solid-State Electrolyte

The optimized sol–gel process was used to fabricate M-doped LLZO ceramics (doping of the Zr^4+^ site; M = Ta or Nb) with a dopant content of 0.2 to 0.6 mol. Ta and Nb substitute at the Zr^4+^ site of LLZO. Figure 2a shows the XRD patterns of the undoped and M-doped LLZO ceramics. The standard XRD data for cubic LLZO (JCPDS 45–109) are plotted at the bottom of the image for comparison. Compared to the standard data for cubic LLZO, peak splitting was observed for each of the XRD peaks of the undoped LLZO, indicating the formation of a tetragonal phase rather than a cubic phase. For the Ta-doped LLZO ceramics, different patterns were observed depending on the dopant content. At 0.2 mol Ta, the patterns changed from tetragonal to a cubic phase, and the two phases were mixed. At 0.4 mol Ta, Ta reacted with the Li_2_O and Li_3_TaO_4_ phases [41]. The Al_2_O_3_ phase was also observed owing to the use of an alumina crucible for the pre-calcination process. At 0.6 mol Ta, the active reaction induced the change into a complete cubic phase, with two secondary phases, MgO and Li_2_CO_3_, introduced from the magnesia crucible during sintering and by the reaction between the pellets and air after the heat treatment, respectively. These phases do not affect the solid-state electrolyte properties [42]. For the Nb-doped LLZO ceramics, a cubic phase was always formed regardless of the dopant content, and no secondary phases were observed other than MgO and Li_2_CO_3_. 

Figure 2b shows FE-SEM images of the fracture surfaces of the M-doped samples. The overall grain size of the M-doped LLZO ceramics was smaller than that of undoped LLZO; however, the grains became more pronounced with an increase in the dopant content in both the Ta- and Nb-doped ceramics, with the tendency of an increase in grain size and uniformity.

Figure 3a,b show the effect of the dopant content of the M-doped LLZO ceramics on the EIS curves and Li-ion conductivity, respectively. The ionic conductivity of the undoped LLZO was too low to be measured using our system; therefore, no data are shown for undoped LLZO. For the Ta-doped LLZO ceramics, the shape of the Nyquist plot changed greatly depending on the dopant content. In contrast, there were no significant changes to the shape of the Nyquist plot with the Nb dopant content. Furthermore, the ionic conductivity—which was much higher for the Nb-doped ceramics than the Ta-doped ceramics—increased slightly with the content of each dopant. The LLZO ceramic doped with 0.6 mol of Nb exhibited the highest ionic conductivity of 1.672 × 10^−4^ S/cm.

The ionic conductivity is improved by Zr^4+^ site doping owing to expansion of the Li^+^ transfer bottleneck. Previous research [43,44] has described this mechanism as widening of the migration channel through which Li^+^ can be conducted, thus making diffusion easier. Furthermore, this widening is greater in the case of Nb doping than Ta doping. Another plausible explanation is based on the XRD results; Nb doping has a stronger tendency than Ta doping to stabilize the cubic phase, leading to an improvement in the ionic conductivity.

The effects of mono-doping at the next cation substitution site, Li^+^, were examined by considering various A-doped LLZO ceramics (doping of the Li^+^ site; A = Al, Ga, or Ge) with a dopant content of 0.2 to 0.6 mol. Figure 4a,b show XRD patterns and FE-SEM images, respectively, of the A-doped LLZO ceramics. Many pores and cracks were generated in the pellets when doping with Al, Ga, and Ge. In particular, multiple secondary phases were observed in the Al-doped ceramic because there was insufficient active reaction between the elements. This result differs considerably from those of a previous study [45], in which Al served as a sintering aid and caused densification of the LLZO and stabilization of the cubic phase, as the processing method was different (solid-state reaction vs. sol–gel method). In the Ga-doped LLZO ceramics, a LiGaO_2_ secondary phase was observed with a very weak Li_2_ZrO_3_ peak caused by excess Li and MgO peak due to the magnesia crucible used for sintering. It has been reported that LiGaO_2_ is formed by a reaction between excess Li and Ga and acts as a sintering agent [46]. Overall, with the increase in the Ga dopant content, the peak intensity of cubic LLZO decreased significantly. 

The fracture surfaces of the Ga-doped ceramics exhibited traces of the liquid phase that forms during sintering by the dissolution of some material. When the doping content was small (0.2 mol Ga), the grains of LLZO were slightly covered by this liquid phase; however, as the Ga content increased, the pores increased or an excessive liquid phase was formed, with almost no clear grain structure. Therefore, excessive Ga doping can have an adverse effect on the crystallinity and sintering stability [47].

Rb was also used as a mono-dopant for substitution at the La^3+^ site; however, no cubic phase of LLZO was formed at any of the tested Rb contents (0.2–0.6 mol).

### 3.2. Co- and Tri-Doped LLZO Solid-State Electrolyte

In the mono-doping experiments, the best ionic conductivity (4.889 × 10^−5^ S/cm) was observed for M-doped LLZO (Zr^4+^ site doping) with 0.6 mol Nb, while the best crystallinity and sintering stability were obtained for A-doped LLZO (Li^+^ site doping) with 0.2 mol Ga. Therefore, these two compositions were introduced as co-dopants to LLZO. In addition, while mono-doping of Al at the Li^+^ site did not produce ceramics with good sintering stability, the EIS measurements revealed that the Al-doped LLZO (0.6 mol Al) had the second highest Li-ion conductivity (2.713 × 10^−5^ S/cm). Thus, co-doping with 0.6 mol Al and 0.6 mol Nb was also performed. While mono-doping with Rb did not yield cubic phase LLZO, we performed co-doping with 0.125 mol Rb and 0.6 mol Nb based on the results of a simulation [48] of the effect of Rb content.

Figure 5a,b show XRD patterns and fracture surfaces, respectively, of the three co-doped LLZO ceramics (doping of two ion sites). The XRD peaks associated with the cubic phase of LLZO became sharper compared to those of the mono-doped ceramics, and the formation of secondary phases due to excess Li was significantly reduced. Furthermore, the fracture surfaces (Figure 5b) demonstrated that the grain distribution was uniform, with fewer pores than the mono-doped ceramics. Overall, a regular grain was observed. Particularly stable and uniform grains were observed for the 0.125 mol Rb/0.6 mol Nb co-doped LLZO. An optimal amount of Rb doping promotes densification of the garnet structure. The ionic radius of Rb^+^ is larger than that of La^3+^ (1.46 vs. 1.06 Å); thus, Rb^+^ doping of the La^3+^ site will expand the Li^+^ ion diffusion pathway, positively affecting the ionic conductivity [31,49].

Based on the co-doped LLZO ceramics with Nb occupying the Zr^4+^ site and Rb occupying the La^3+^ site, tri-doping was conducted with Ga or Al occupying the Li^+^ site. Figure 6a,b show the XRD patterns and fracture surfaces, respectively, of the tri-doped LLZO ceramics. The XRD patterns confirmed that tri-doping with Nb, Rb, and Ga or Al resulted in the formation of a pure cubic phase with almost no secondary phases. As shown in Figure 6b, the grains were clear, with traces of the liquid phase at the grain boundaries.

Figure 7 shows Nyquist plots of the co- and tri-doped LLZO ceramics based on EIS measurements. The plots of the co-doped and tri-doped ceramics can be clearly distinguished. For the co-doped samples, the impedance tends to decrease with a constant slope as the frequency increases, whereas for the tri-doped samples, the impedance slowly decreases as the frequency increases, with a sharp decrease of the imaginary part of impedance at a low real impedance.

The Li-ion conductivities of the co- and tri-doped LLZO ceramics were calculated using the equivalent circuit and summarized in Table 1. Compared with the value of 1.672 × 10^–4^ S/cm for Nb-doped LLZO (mono-doped), the ionic conductivity was improved by co-doping with Rb or Al; however, co-doping with Ga slightly decreased the ionic conductivity. Nevertheless, when tri-doping was performed with Ga, Rb, and Nb, an increase in the ionic conductivity by a factor of nearly 1.5 was observed compared to that of Ga,Nb-doped LLZO.

### 3.3. Effect of Rb Content on Properties of Tri-Doped LLZO Ceramics

The addition of Rb to the co- and tri-doped ceramics appeared to improve the ionic conductivity of LLZO. Therefore, we hypothesized that Rb plays a key role in the ionic conductivity of Li^+^. To determine the effect of the Rb content on the properties of tri-doped LLZO, we fabricated tri-doped LLZO samples with Rb contents ranging from 0 (0.2 mol Ga/0.6 mol Nb co-doped sample) to 0.15 mol. We used the Ga,Rb,Nb-doped sample, which exhibited the best properties in the tri-doping experiments in Section 3.2. Figure 8a,b show XRD patterns and fracture surfaces, respectively, of the tri-doped LLZO ceramics with different Rb contents. For all Rb contents, cubic phase LLZO was formed with almost no impurities. The secondary phases generated were mainly LiGaO_2_, which has a positive effect on sintering, and Li_2_CO_3_ and Rb_2_CO_3_—with small peak intensities—which do not compromise the Li-ion conductivity. From the fracture surface images in Figure 8b, it was revealed that, overall, the porosity and grain size increased as the Rb content increased.

Figure 9 shows the Li-ion conductivity, relative density, shrinkage, and activation energy of tri-doped LLZO ceramic according to the Rb content. In comparison with the ceramic with 0 mol Rb, a 0.1 mol Rb addition increased the ionic conductivity, shrinkage, and relative density. This is because an increase in the shrinkage and relative density decreases the porosity, which creates a more compact morphology at the grain boundaries and increases the ionic conductivity. In addition, the activation energy for Li-ion conduction was reduced, which is thought to be because the fraction of pores causing bottlenecks for Li ions was reduced, and intergranular Li^+^ migration (between grains) was facilitated [50]. When the Rb content was increased further, the shrinkage and relative density decreased, because the porosity increased, as shown in Figure 8. Therefore, the ionic conductivity of Li was decreased, while the activation energy for ion conduction increased. From these experiments, the best Li-ion conductivity at room temperature and lowest activation energy obtained were 2.222 × 10^−4^ S/cm and 0.23 eV, respectively, for the tri-doped LLZO with 0.1 mol Rb, 0.2 mol Ga, and 0.6 mol Nb.

## 4. Conclusions

In this study, LLZO solid-state electrolyte ceramics were fabricated using a sol–gel synthesis technique for substitutional doping at the Li^+^, La^3+^, and Zr^4+^ sites of LLZO. The effect of the combination and content of dopants on the properties of LLZO solid-state electrolyte was investigated. LLZO ceramics with a stable cubic phase and relatively high Li-ion conductivity were obtained through co- and tri-doping. The dopants Ta and Nb, which substituted at the Zr^4+^ site, had a stabilizing effect on the cubic phase, while Nb also greatly improved the Li-ion conductivity. Al and Ge dopants, which substituted at the Li^+^ site, showed unstable properties when introduced as mono-dopants, but the addition of a small amount of Ga improved the sintering stability. Rb dopant, which substituted at the La^3+^ site, did not produce a cubic phase when used as a mono-dopant; however, a densification effect was achieved when co-doping with Nb. The co-doped LLZO ceramics showed a significant decrease in the number and intensity of secondary phases, and the sintering properties were improved compared to the mono-doped ceramics. By designing a reasonable tri-doping composition based on the results of the co-doping experiments, i.e., tri-doping with Nb, Ga, and Rb, a pure cubic phase with almost no secondary phase, uniform grain structure, and excellent Li-ion conductivity was obtained. By tailoring the Rb content, the shrinkage and relative density were improved, with the best results observed for the ceramic with 0.1 mol Rb, 0.2 mol Ga, and 0.6 mol Nb; a high Li-ion conductivity of 2.222 × 10^−4^ S/cm at room temperature and low activation energy of 0.23 eV were achieved.

## Figures and Tables

**Figure 1 micromachines-12-00134-f001:**
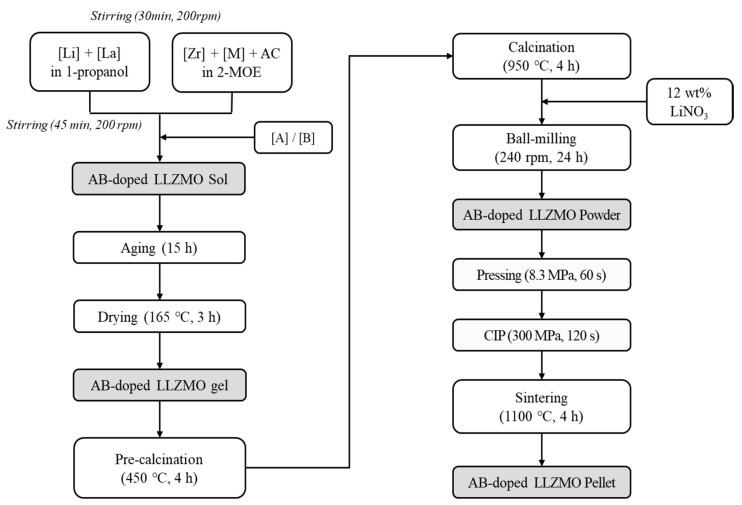
Optimized synthesis process of doped- Li_7_La_3_Zr_2_O_12_ (LLZO) ceramics (AC: acetic acid solution; 2-MOE: 2-methoxyethanol; M: Zr^4+^ site dopant; A: Li^+^ site dopant; B: La^3+^ site dopant; CIP: cold isostatic pressing; LLZMO: M-doped LLZO).

**Figure 2 micromachines-12-00134-f002:**
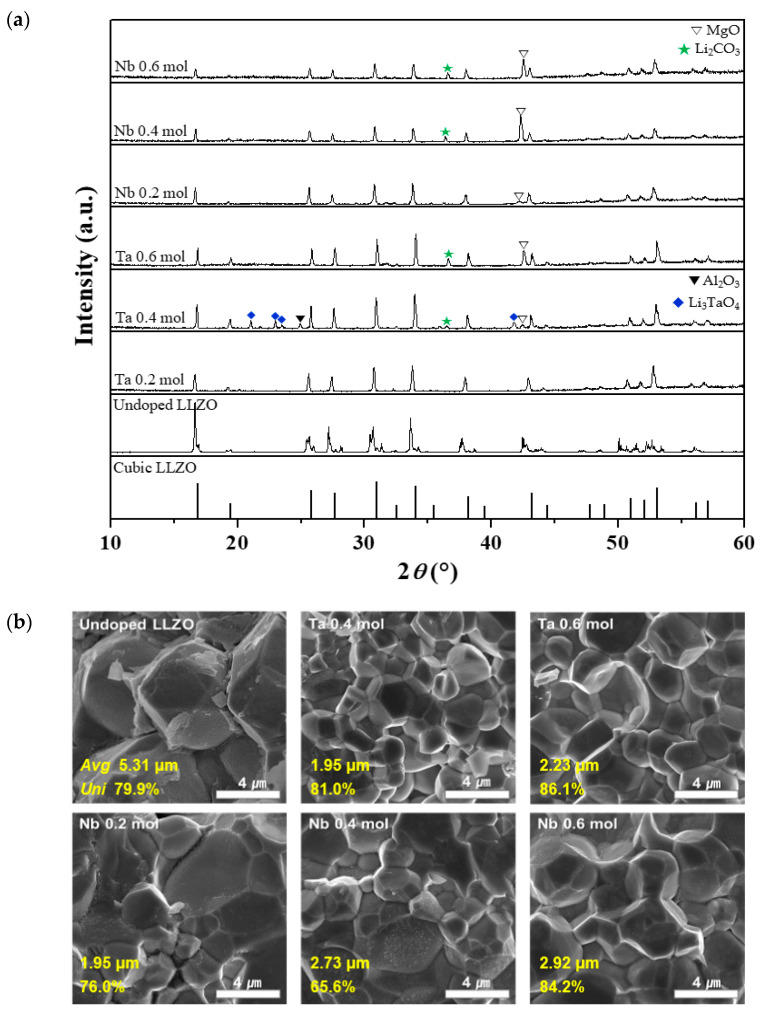
Crystal structure and morphology of M-doped LLZO ceramics (mono-doping of the Zr^4+^ site with 0.2–0.6 mol Ta or Nb): (**a**) X-ray diffraction (XRD) patterns and (**b**) field-emission scanning electron microscopy (FE-SEM) images of the fracture surfaces (*Avg* and *Uni*: the average and the uniformity of grain size, respectively).

**Figure 3 micromachines-12-00134-f003:**
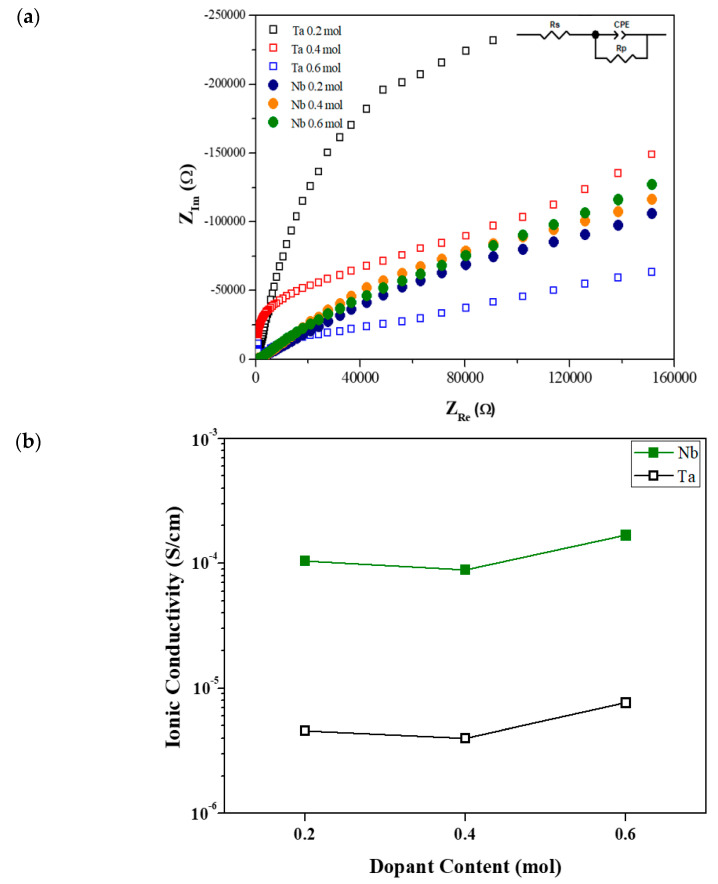
Electrochemical impedance spectroscopy (EIS) results of M-doped LLZO ceramics (mono-doping of the Zr^4+^ site with 0.2–0.6 mol Nb or Ta): (**a**) Nyquist plots (*Z*_Re_ and *Z*_Im_: real and imaginary part of impedance, respectively; inset: equivalent circuit, where *R*_s_ is electrolyte resistance; *R*_p_ is resistance due to charge accumulation at the interface; and CPE is constant phase element); (**b**) Li-ion conductivity according to dopant concentration.

**Figure 4 micromachines-12-00134-f004:**
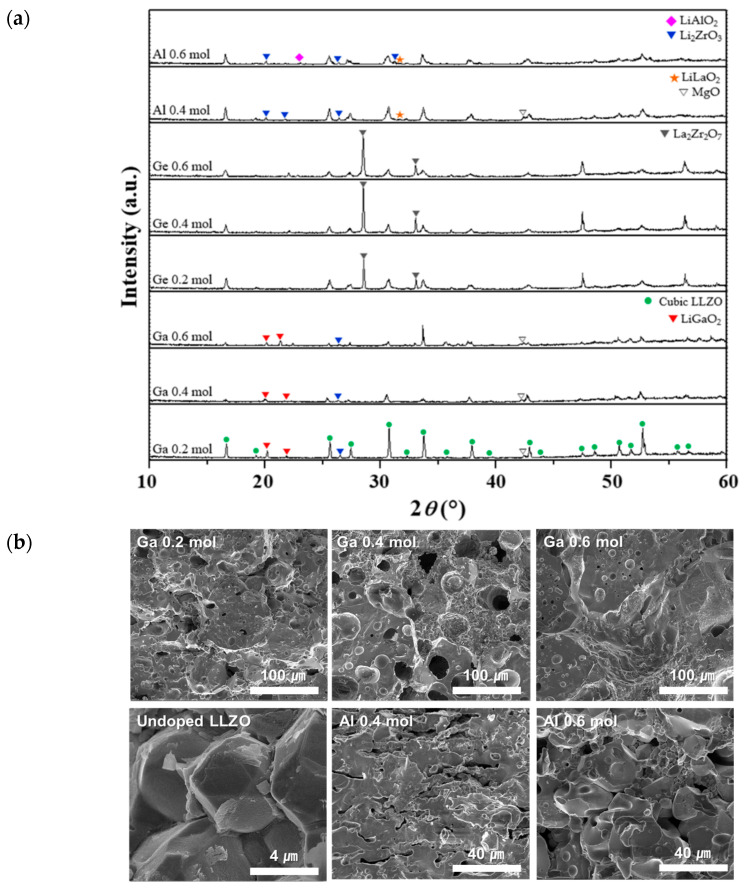
Crystal structure and morphology of A-doped LLZO ceramics (mono-doping of the Li^+^ site with 0.2–0.6 mol Al or Ge): (**a**) XRD patterns and (**b**) FE-SEM images of the fracture surfaces.

**Figure 5 micromachines-12-00134-f005:**
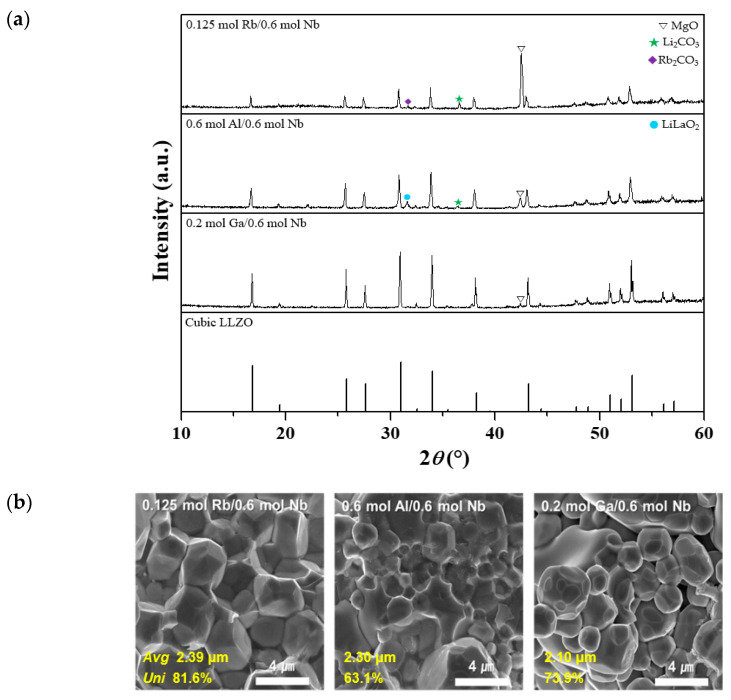
Crystal structure and morphology of co-doped LLZO ceramics (co-doping of 0.125 mol Rb, 0.6 mol Al, or 0.2 mol Ga with 0.6 mol Nb): (**a**) XRD patterns and (**b**) SEM images of the fracture surfaces (*Avg* and *Uni*: the average and the uniformity of grain size, respectively).

**Figure 6 micromachines-12-00134-f006:**
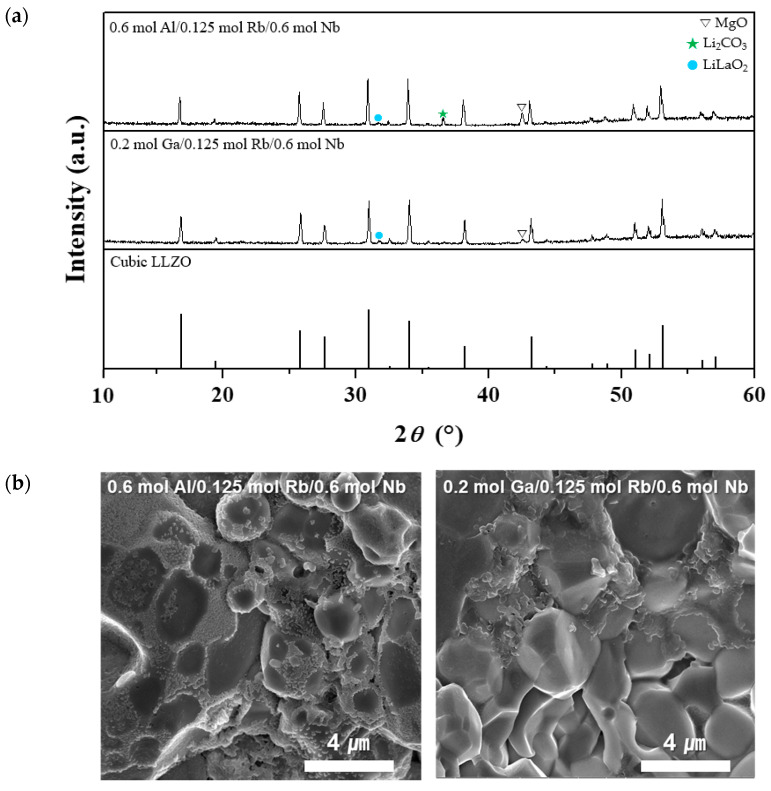
Crystal structure and morphology of tri-doped LLZO ceramics (tri-doping of 0.6 mol Al or 0.2 mol Ga with 0.125 mol Rb and 0.6 mol Nb): (**a**) XRD patterns and (**b**) FE-SEM images of the fracture surfaces.

**Figure 7 micromachines-12-00134-f007:**
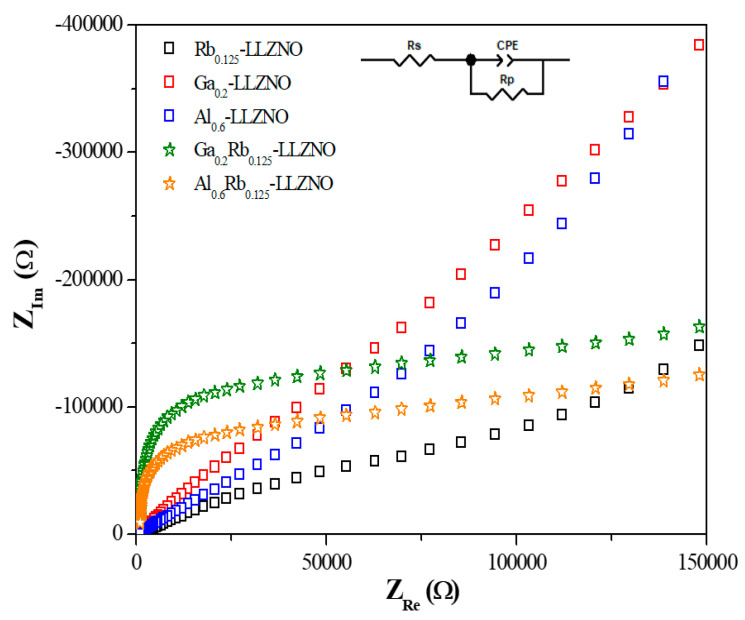
Nyquist plots of co- and tri-doped LLZO solid-state electrolyte ceramics (*Z*_Re_ and *Z*_Im_: real and imaginary part of impedance, respectively).

**Figure 8 micromachines-12-00134-f008:**
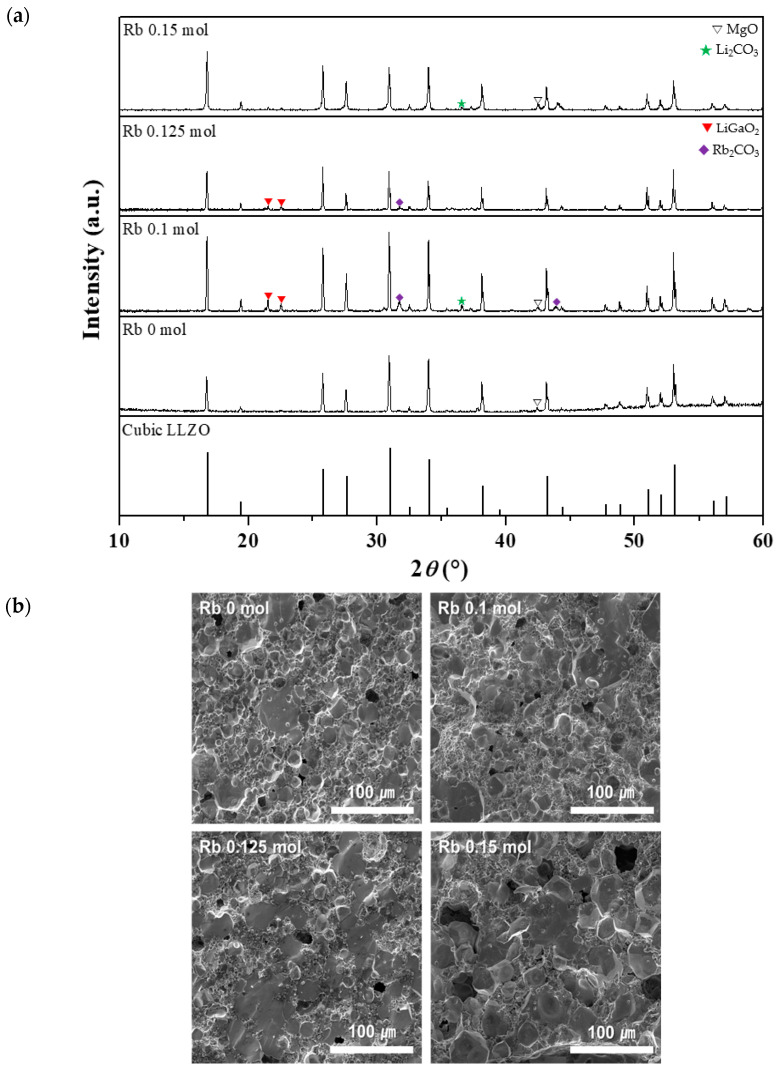
Crystal structure and morphology of tri-doped LLZO ceramics with 0.2 mol Ga, 0.6 mol Nb, and 0–0.15 mol Rb: (**a**) XRD patterns and (**b**) FE-SEM images of fracture surfaces.

**Figure 9 micromachines-12-00134-f009:**
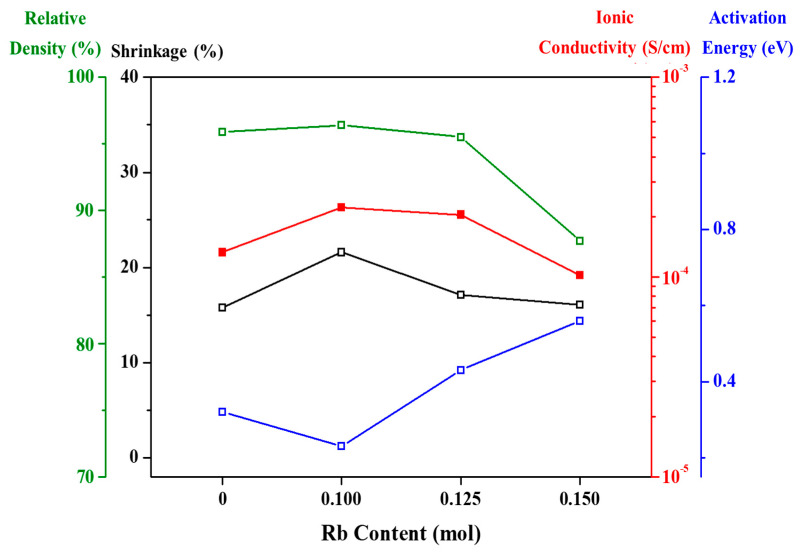
Changes in physical properties including Li-ion conductivity of tri-doped LLZO ceramics according to Rb content.

**Table 1 micromachines-12-00134-t001:** Li-ion conductivity of co- and tri-doped LLZO solid-state electrolyte ceramics.

Sample	Dopants	Ionic Conductivity (S/cm)
Rb_0.125_Li_7.44_La_3_Zr_1.4_Nb_0.6_	0.125 mol Rb/0.6 mol Nb	2.129 × 10^−4^
Al_0.6_Li_6.84_La_3_Zr_1.4_Nb_0.6_	0.6 mol Al/0.6 mol Nb	2.024 × 10^−4^
Ga_0.2_Li_7.24_La_3_Zr_1.4_Nb_0.6_	0.2 mol Ga/0.6 mol Nb	1.330 × 10^−4^
Ga_0.2_Rb_0.125_Li_7.24_La_3_Zr_1.4_Nb_0.6_	0.2 mol Ga/0.125 mol Rb/0.6 mol Nb	2.053 × 10^−4^
Al_0.6_Rb_0.125_Li_6.84_La_3_Zr_1.4_Nb_0.6_	0.6 mol Al/0.125 mol Rb/0.6 mol Nb	3.956 × 10^−5^

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
