# Peer review of "Tri-Doping of Sol–Gel Synthesized Garnet-Type Oxide Solid-State Electrolyte"

_micromachines, 2021, doi:10.3390/mi12020134_

Round 1

Reviewer 1 Report

Dear Authors,

I found your article very interesting but i have few questions:

  1. Could you mention the ionic conductivity of LLZO and then compare with your doped materials? how much percentage did you improve?
  2. Why the sol-gel is better than solid state reaction. In your case, you are also using a very high sintering temperature for very long. 
  3. Rb, Ga and Nb, you reported that high lithium ion conductivity but undoped LLZO also shows the ionic conductivity in the same range? Could you mention, what are the benefits of doping?
  4. What was the thickness of the pellet? How much weight the pellet looses upon sintering, why?
  5. You used R(RQ) circuit to fit your data. could you explain it? Could you put the resistance table e.g. grain boundary resistance table with respect to doped samples.

Thank you

Author Response

Dear reviewer,

Authors appreciate your valuable comments.

Our manuscript has been revised and improved according to your professional reviews.

Thank you again.

Best regards,

Reviewer 2 Report

From line 150-153, the author used one sem image to indicate the grain size. however it might not be sufficiently representative and statistical. could the author please provide additional data on grain size vs dopent content and more statistical? additionally, how could the author define the uniformity of grain? the author needs to specify or use the concept of "roundness" to define. otherwise it is hard to draw this conclusion.

from line 178-180, the author claims that "

Many pores and cracks were generated in the pellets when doping with Al, Ga, and Ge, excluding the case with 0.2 mol.'

However, it seems the existence of pores are also obvious in 0.2mol Ga, but 0.6mol Ga has less pores. Please be precise in describing these experimental results.

I suggest the author to use statistical data to indicate grain size, otherwise it is not convincing to me. 

Author Response

(The authors gave the same response as above.)
